# A Combination of Glutaminase Inhibitor 968 and PD-L1 Blockade Boosts the Immune Response against Ovarian Cancer

**DOI:** 10.3390/biom11121749

**Published:** 2021-11-23

**Authors:** Jing-Jing Wang, Michelle Kwan-Yee Siu, Yu-Xin Jiang, Thomas Ho-Yin Leung, David Wai Chan, Huo-Gang Wang, Hextan Yuen-Sheung Ngan, Karen Kar-Loen Chan

**Affiliations:** Departments of Obstetrics and Gynaecology, University of Hong Kong, Hong Kong, China; wjj01947@163.com (J.-J.W.); mkysiu@gmail.com (M.K.-Y.S.); yuxin_jiang2012@126.com (Y.-X.J.); thomashyleung@gmail.com (T.H.-Y.L.); dwchan@hku.hk (D.W.C.); wanghuog@connect.hku.hk (H.-G.W.); hysngan@hku.hk (H.Y.-S.N.)

**Keywords:** ovarian cancer, immunotherapy, T cells, glutamine metabolism

## Abstract

Programmed cell death 1 ligand (PD-L1) blockade has been used therapeutically in the treatment of ovarian cancer, and potential combination treatment approaches are under investigation to improve the treatment response rate. The increased dependence on glutamine is widely observed in various type of tumors, including ovarian cancer. Kidney-type glutaminase (GLS), as one of the isotypes of glutaminase, is found to promote tumorigenesis. Here, we have demonstrated that the combined treatment with GLS inhibitor 968 and PD-L1 blockade enhances the immune response against ovarian cancer. Survival analysis using the Kaplan–Meier plotter dataset from ovarian cancer patients revealed that the expression level of GLS predicts poor survival and correlates with the immunosuppressive microenvironment of ovarian cancer. 968 inhibits the proliferation of ovarian cancer cells and enhances granzyme B secretion by CD8^+^ T cells as detected by XTT assay and flow cytometry, respectively. Furthermore, 968 enhances the apoptosis-inducing ability of CD8^+^ T cells toward cancer cells and improves the treatment effect of anti-PD-L1 in treating ovarian cancer as assessed by Annexin V apoptosis assay. In vivo studies demonstrated the prolonged overall survival upon combined treatment of 968 with anti-PD-L1 accompanied by increased granzyme B secretion by CD4^+^ and CD8^+^ T cells isolated from ovarian tumor xenografts. Additionally, 968 increases the infiltration of CD3^+^ T cells into tumors, possibly through enhancing the secretion of CXCL10 and CXCL11 by tumor cells. In conclusion, our findings provide a novel insight into ovarian cancer cells influence the immune system in the tumor microenvironment and highlight the potential clinical implication of combination of immune checkpoints with GLS inhibitor 968 in treating ovarian cancer.

## 1. Introduction

Interest in boosting the immune system to fight cancer cells was evident in the literature as early as the late nineteenth century [1]. Since then, five key immunotherapy modalities have become cutting-edge cancer treatments after undergoing extensive research and clinical trials [2]. Among them, the immune checkpoint inhibitors function to block immune checkpoint receptors, thereby reactivating tumor-infiltrating lymphocytes to perform tumor elimination functions. Immune checkpoint inhibitors, including monoclonal antibodies targeting cytotoxic T lymphocyte-associated antigen 4 (CTLA4), programmed cell death 1 (PD1), or PD1 ligand (PD-L1), are effective in the treatment of various types of cancer, especially melanoma and lung cancer [1]. In ovarian cancer, however, there has been no approved immune checkpoint blockade therapy proposed to date, even though promising results were reported in initial trials. 

Currently, the first-line treatment for advanced ovarian cancer is debulking surgery followed by neoadjuvant chemotherapy. However, most patients eventually suffer from relapse regardless of a good response to initial treatment. Lymphocyte infiltration in ovarian cancer and the positive correlation between lymphocyte infiltration and clinical outcomes suggested that the immune system plays an essential role in the development of ovarian cancer [3]. Studies have also reported an inverse correlation between the aberrant expression of PD-L1 on tumor cells and patient prognosis in many human tumors including ovarian cancer [4,5]. Besides, the overall survival of C57BL/6 mice bearing ID8 tumors treated with anti-PD-L1 antibody was shorter than those treated with carboplatin [6] or paclitaxel [7] alone. In such cases, immunotherapy seems to be a promising treatment strategy to improve the clinical outcomes of ovarian cancer. Hence, numerous studies are underway to confirm the clinical efficacy of immune checkpoint therapy on ovarian cancer. However, remission was only observed in about 10–15% of patients with advanced and recurrent ovarian cancer following treatment with immune checkpoint inhibitors [8,9]. Given these modest response rates, new approaches are needed to explore the possible agents that could cooperate with these immune checkpoint inhibitors to improve the clinical response.

The rewiring of cellular metabolism has become an essential hallmark of cancer since the discovery of aerobic glycolysis by Otto Warburg in 1924. As well as glycolysis, glutamine reprogramming has also gained attention as an important target for cancer. Glutamine is the most abundant non-essential amino acid and provides precursors for the biosynthesis of other amino acids, proteins, nucleotides, and other molecules [10]. An increased requirement for glutamine in lung cancer [11], renal cell carcinoma [12], chronic lymphocytic leukemia [13], and ovarian cancer [14] to fulfill their metabolic needs has been reported. In ovarian cancer, glutamine dependence was found to correlate with cancer invasiveness and patient survival [14]. Moreover, a recent study using an in vivo model of colon cancer revealed that blockade of glutamine modulating the metabolism of cancer cells and immune cells could serve as a metabolic checkpoint for cancer immunotherapy [15]. 

As the first enzyme in the glutaminolysis process, glutaminase converts glutamine into glutamate, and further to α-ketoglutarate (α-KG) in the mitochondria, and two key isotypes of glutaminase, the kidney-type glutaminase (GLS) and the liver-type glutaminase, have been identified in humans [16]. GLS attracted considerable attention as a predictor of the prognosis of cancer, and the implications of its inhibition on cancer biology has become an area deserving active investigation. The chemical compound 5-(3-bromo-4-(dimethylamino)phenyl)-2,2-dimethyl-2,3,5,6-tetrahydrobenzo[a]phenanthridin-4(1*H*)-one, designated as compound 968, was discovered to be an allosteric inhibitor of GLS, and its inhibitory potential on cancer cell migration and proliferation have been reported [17,18]. In ovarian cancer, compound 968 significantly inhibited cell proliferation and increased the sensitivity of cells to treatment with paclitaxel [19].

In this study, we provide evidence of increased T cell function, as well as the elevated secretion of C-X-C motif chemokine 10 (CXCL10) and C-X-C motif chemokine 11 (CXCL11) by cancer cells, in response to exposure to compound 968. Furthermore, the improved overall survival of ID8 mice after co-treatment with compound 968 and anti-PD-L1 antibody and the higher proportion of granzyme B-expressing CD4^+^ and CD8^+^ T cells isolated from tumors from the co-treatment group provided the rationale for further experiment to confirm the efficacy and safety of this combined treatment in ovarian cancer. Compound 968 was also shown to increase CD3^+^ T cell infiltration into tumors. Our collective findings revealed that the increased T cell function and infiltration into tumors in response to compound 968, and the inhibition of glutaminase by compound 968, helped to improve the treatment effect of immune checkpoint blockade in ovarian cancer. 

## 2. Materials and Methods

### 2.1. Cell Lines and Culture

Human ovarian cancer cell lines including A2780CP, ES2, OVCA433, and SK-OV-3, as well as the mice ovarian cancer cell line ID8, were used in this study. The A2780CP cell line was provided by Prof. Benjamin Tsang (Department of Obstetrics & Gynecology, University of Ottawa, (Ottawa, ON, Canada). The OVCA433 cell line was provided by Prof. George Tsao (School of Biomedical Sciences, LKS Faculty of Medicine, The University of Hong Kong, Hong Kong). The ID8 cell line was a gift from Dr. Katherine Roby (Department of Anatomy and Cell Biology, University of Kansas Medical Center, Kansas, KS, USA). The other cell lines were purchased from the American Type Culture Collection (Manassas, VA, USA). Among them, A2780CP, OVCA433, SK-OV-3, and ID8 cells were maintained in DMEM (#11965118, Invitrogen, San Diego, CA, USA), and ES2 cells were maintained in RPMI (#11875119, Invitrogen), with 10% fetal bovine serum (FBS, #10270106, Invitrogen) and 100 U/mL penicillin-streptomycin (#15140122, Invitrogen) added to the media. All cells were cultured in a humidified incubator at 37 °C and 5% CO_2_.

### 2.2. Purified CD8^+^ T Cell Populations

Peripheral blood mononuclear cells (PBMCs) were separated by centrifugation in lymphoprep^TM^ density gradient medium (#07861, StemCell Technologies, Vancouver, BC, Canada) from buffy coat leukocyte concentrates obtained from healthy female donors provided by the Hong Kong Red Cross. CD8^+^ T cells were purified by incubating the PBMCs with RosetteSep human CD8^+^ T cell Enrichment Cocktail (#15063, StemCell Technologies) before gradient centrifugation according to the instructions recommended by the supplier. For CD8^+^ T cell culture, complete RPMI medium containing 10% FBS and 100 U/mL penicillin-streptomycin was used. 

### 2.3. T-Cell/Ovarian Cancer Cell Co-Culture System

Similar to the coculture system reported by Xu et al. [20], we stimulated PBMCs with anti-CD3e (2.5 μg/mL, #300432, BioLegend, San Diego, CA, USA), anti-CD28 (1.25 μg/mL) (#302923, BioLegend) and ovarian cancer cell lysate. Briefly, PBMCs were plated at a density of 2 × 10^6^ per well in six-well plates and stimulated with anti-CD3e, anti-CD28, as well as ovarian cancer cell lysate for 3 days. Then CD8^+^ T cells were purified using RosetteSep human CD8^+^ T cell Enrichment Cocktail. For the direct co-culture system, activated CD8^+^ T cells were co-cultured with ovarian cancer cells in 12-well plates at a ratio of 5:1 [21,22] in the presence of different doses of glutaminase inhibitor 968, with or without anti-PD-L1 antibody. IgG was used as a control. After 24 h, the CD8^+^ T cells were removed with phosphate-buffered saline (PBS), and the attached cancer cells were collected and subjected to flow cytometry. For the indirect co-culture system, transwell cell culture chambers (0.4 μm, CLS3397, Corning, Tewksbury, MA, USA) were used to separate CD8^+^ T cells and cancer cells. CD8^+^ T cells were seeded into the upper chamber, and cancer cells were seeded into the lower chamber at a ratio of 5:1. After exposure to different concentrations of 968 for 24 h, the cancer cells were collected for flow cytometry analysis.

### 2.4. Quantitative Real-Time PCR

Total RNA from cancer cells was isolated by a NucleoSpin RNA kit (#740955, Macherery-Nagel, Düren, Germany) according to the manufacturer’s instructions. We used the SuperScript VILO^TM^ Master Mix (#11755250, Invitrogen) to synthesize cDNA. Then, quantitative real-time PCR was carried out to quantify target gene expression. The sequences of the gene-specific primers used in this experiment were as follows: CXCL10 (Forward 5′-CCAGAATCGAAGGCCATCAA-3′; Reverse 5′-CATTTCC TTGCTAACTGCTTTCAG-3′), CXCL11 (Forward 5′-AGAGGACGCTGTCTTTGCAT-3′; Reverse 5′-TGGGATTTAGGCATCGTTGT-3′), GAPDH (Forward 5′-TCCATGACAAC TTTGGTATCGTG-3′; Reverse 5′-ACAGTCTTCTGGGTGGCAGTG-3′). GAPDH was used as the reference gene.

### 2.5. 2,3-Bis-(2-methoxy-4-nitro-5-sulfophenyl)-2h-tetrazolium-5-carboxanilide (XTT) Assay

Cell viability following exposure to glutaminase inhibitor 968 was determined by an XTT assay kit (Roche, Brighton, MA, USA), according to the manufacturer’s instructions. The cancer cells were seeded in 96-well plates at a density of 1000 or 2000 cells/well and were treated with increasing doses of 968 for the indicated times at 37 °C and 5% CO_2_. Doses we used were based on a previous study [19]. The medium was then replaced with the XTT labeling mixture and incubated for another 4 h before measuring the absorbance. The optical density was measured by an ELISA plate reader (INFINITE F NANO^+^, TECAN, Männedorf, Switzerland) at 495 nm with a reference wavelength of 650 nm. Each well was examined in triplicate, and three independent experiments were performed.

### 2.6. Cancer Cell Apoptosis Assay

After treatment with different doses of 968, with or without anti-PD-L1 antibody, the cancer cells harvested from the co-culture system were subjected to flow cytometry to determine the percentage of apoptotic cells. Briefly, cells were collected, counted, and 2 × 10^5^ cells were added to assay tubes and subsequently washed with PBS. Cells were then stained with PE-anti-CD8 antibody (#555635, BD Pharmingen, San Diego, CA, USA) for 30 min at 4 °C, protected from the light. After centrifugation and rinsing, cells were resuspended in 1 × binding buffer containing annexin V (BD Pharmingen). After incubation for 15 min at room temperature in the dark, the stained cells were immediately analyzed on a BD LSR Fortessa analyzer and the flow cytometry results were analyzed by FlowJo software. The annexin V (+) cells under the CD8 (−) gate were considered to be apoptotic cancer cells [23]. For indirect co-culture system, cancer cells were collected, counted, and stained with annexin V at the end of coculturing and then analyzed by BD LSR Fortessa analyzer. 

### 2.7. In Vivo Experiment 

Immunocompetent C57BL/6J mice were obtained from the Laboratory Animal Unit of the University of Hong Kong. For the in vivo experiments, 5 × 10^6^ ID8 cells in 200 μL PBS were transplanted via intraperitoneal injection into 4–6-week-old female C57BL/6J mice. Then, 28 days post tumor implantation, mice were randomly divided into four groups and administered: (1) compound 968 + an isotype of anti-PD-L1 antibody (LEAF^TM^, #400637); (2) compound 968 + anti-PD-L1 (LEAF^TM^, #124309); (3) vehicle + anti-PD-L1; or (4) vehicle + an isotype of anti-PD-L1 via intraperitoneal injection. Compound 968 was injected at a dose of 200 μg per mouse daily for 14 days, and anti-PD-L1 antibody was injected at a dose of 200 μg per mouse daily for 5 days. Mice were monitored daily for body weight and symptoms such as depressed lethargic inactive or non-ambulatory, abnormal breathing, hunched back or ruffled fur, pale extremities, abdominal distension, muscle atrophy, emaciated, hypothermic or dehydrated. They were euthanized on the appearance of body weight loss >20% of original body weight or appearance of four symptom on three consecutive days. The animal experiment in this study was approved by the University of Hong Kong Committee on the Use of Live Animals in Teaching and Research (CULATR No. 5599-20) and performed following the Animals (Control of Experiments) Ordinance (Hong Kong) and the Institutes’s guidance on animal experiments.

### 2.8. Flow Cytometry Analysis 

Granzyme B secretion by CD8^+^ T cells after treatment with compound 968 was examined by flow cytometry. Briefly, CD8^+^ T cells were collected after exposure to different doses of 968 for 48 h, and then washed and placed into an assay tube. After staining with surface markers, FITC anti-human CD8 or isotype control (BD Pharmigen, San Diego, CA, USA), the cells were permeabilized with Cytofix/cytoperm (BD Pharmigen). The cells were then incubated with PE anti-human granzyme B or the isotype control for 30 min at 4 °C protected from the light. 

Tumor nodules harvested from the mice were dissociated using the Minute^TM^ cell suspension isolation kit (#SC-012, Invent, Plymouth, MN, USA) following the manufacturer’s instructions. After filtering through a 40-μm cell strainer and lysing the red blood cells using ACK lysis buffer (#A1049201, Invitrogen), the suspended cells were separated by centrifugation in Percoll gradient (#17-0891-01, Sigma-Aldrich, St. Louis, MO, USA) medium. Thereafter, the enriched mononuclear cells were stained with BV510-anti-CD3e (BD, #563024), FITC-anti-CD4 (#553729, eBioscience^TM^, San Diego, CA, USA), or PE-anti-CD8A according to the protocols provided by the suppliers. To stain intracellular granzyme B, cells were fixed, permeabilized, and then incubated with Alexa Fluor^®^ 647 anti-granzyme B for 30 min. Samples were run on a BD LSR Fortessa analyzer and analyzed by FlowJo software.

### 2.9. Kaplan–Meier Plotter and Gene Expression Profiling Interactive Analysis (GEPIA) 

Kaplan–Meier survival analysis of GLS in ovarian cancer was performed using the Kaplan–Meier plotter dataset from ovarian cancer patients including all tumor stages, grades and histological subtypes, (http://kmplot.com/analysis/index.php?p=service&cancer=ovar) (accessed date 30 Jaunary 2019), and the cut-off value was the median of the GLS mRNA expression level. The number of ovarian cancer patients used for overall and progress-free survival analysis were 1656 and 1435, respectively. The correlation between GLS and immunosuppressive genes was based on GEPIA (http://gepia.cancer-pku.cn/) (accessed date 30 Jaunary 2019) using Spearman’s rank correlation coefficient [24]. 

### 2.10. Statistical Analysis

Data were analyzed by GraphPad Prism 6 (GraphPad Software Inc., San Diego, CA, USA). Representative results from three independent experiments are presented. Comparisons between two groups were based on the Student’s t-test, and data between three groups were compared using one-way ANOVA. For survival analysis, the log-rank test was used. *p* values < 0.05 were considered significant.

## 3. Results

### 3.1. GLS Upregulation Predicts Poor Survival and Correlates with an Immunosuppressive Microenvironment in Ovarian Cancer

First, we used the Kaplan–Meier plotter dataset from ovarian cancer patients (including all tumor stages, grades and histological subtypes) to analyze the predictive value of GLS in ovarian cancer, and the log-rank test results indicated a negative correlation between GLS expression and overall survival (*n* = 1656; Figure 1A) and progression-free survival (*n* = 1435; Figure 1B).

By analyzing the TCGA ovarian cancer dataset generated by GEPIA, a significant albeit very weak correlation was observed between high expression of GLS and upregulation of immunosuppressive genes including PD-L1 (CD274), PD-1 (PDCD1), PD-L2 (PDCD1LG2), CTLA4, and LAG3 (Appendix A), suggesting the potential effects of GLS on immunosuppressive microenvironment in ovarian cancer. To this end, the following functional studies were performed in order to verify such potential effects. 

### 3.2. Compound 968 Inhibits the Proliferation of Ovarian Cancer Cells and Increases the Granzyme B-Secretion by CD8^+^ T-Cells from Pbmcs 

To determine the effect of compound 968 on the proliferation ability of ovarian cancer cells, an XTT assay was performed on human ovarian cancer cells (ES2, A2780CP, OVCA433 and SKOV-3) after treatment with various doses (5 μM, 10 μM, 25 μM, or 40 μM) of compound 968 for 1 to 3 days. As shown in Figure 2A–E, compound 968 inhibited the proliferation of all cells at a dose of 10 μM for 3 days. To ensure that this effect was not limited to human ovarian cancer cells, we used mouse-derived ovarian cancer cells, ID8, to repeat this assay, and compound 968 exerted a similar suppressive effect on the proliferation ability of these cells at a minimal dose of 5 μM for 3 days.

As a pro-apoptotic serine proteinase, granzyme B, which is primarily secreted by activated cytotoxic T cells and natural killer cells, plays a fundamental role in mediating the eradication of infected cells, allogenic cells, and tumor cells [25]. It has been reported that the number of granzyme B-expressing cells present in colorectal cancer tissue correlates with the prolonged survival of patients [26]. Furthermore, the treatment effects of granzyme B-based cytolytic fusion proteins have been identified in various types of cancer including ovarian carcinoma [27]. In this study, granzyme B secretion by activated CD8^+^ T cells was analyzed by flow cytometry after treatment with different doses of compound 968 (0, 1.5, 2.5, 5, and 10 μM) for 48 h. As shown in Figure 2F,G, compound 968 increased the granzyme B secretion by CD8^+^ T cells in a dose dependent manner. These results indicated that compound 968 could inhibit the proliferation of cancer cell and enhance the granzyme B secretion by CD8^+^ T cells.

### 3.3. Apoptosis of Cancer Cells Co-Cultured with CD8^+^ T-Cells Increases Following Exposure to Compound 968

As granzyme B plays an important role in mediating the apoptosis of cancer cells, we next investigated the apoptosis of cancer cells in the direct and indirect co-culture systems with CD8^+^ T cells at a ratio of 1:5 with or without compound 968 using the annexin V apoptosis assay. As shown in Figure 3, compound 968 was able to induce apoptosis in all three ovarian cancer cell lines at different doses. CD8^+^ T cells that were directly (Figure 3A) or indirectly (Figure 3B) co-cultured with ES2, A2780CP or OVCA433 cells suppressed the survival of these cancer cells (*p* < 0.05). We further observed that the percentage of apoptotic cancer cells significantly increased after direct (Figure 3A) or indirect (Figure 3B) co-culturing with CD8^+^ T cells in the presence of compound 968 compared with other groups (*p* < 0.05). These results suggested that compound 968 helped to increase the apoptosis-inducing activity of CD8^+^ T cells on ovarian cancer cells by inducing granzyme B secretion.

### 3.4. Combination of Compound 968 and Anti-PD-L1 Antibody Enhances The Anticancer Activity of CD8^+^ T Cells

The upregulation of PD-L1 in different cancer cells was either modulated by the oncogenic signaling pathway or triggered by cytokines present in the tumor microenvironment [28,29]. One study reported that the interaction of PD-1, which is typically expressed by T cells, with PD-L1 dampens the cytolytic activity of T cells [30]. Treatments targeting PD1 or PD-L1 have shown breakthrough curative antitumor effects in various types of cancer, including ovarian carcinoma, although only a limited number of patients appear to benefit from the immune checkpoint blockage method [31]. To find an appropriate combinatorial approach to improve the treatment effects of immune checkpoint therapy, we analyzed the effects of combining compound 968 with PD-L1 antibody in ovarian cancer. The functional assay results showed that the percentage of apoptotic ES2 cells in the combination treatment group (compound 968 and anti-PD-L1 antibody) increased by about 15% compared with the group treated with compound 968 alone (compound 968 and IgG), and by approximately 40% compared with the group treated with anti-PD-L1 antibody alone (Figure 3C). This finding confirmed that the combination of compound 968 with PD-L1 antibody could be an effective co-treatment method to improve the clinical effects of immunotherapy.

### 3.5. Combination of Compound 968 and Anti-PD-L1 Antibody Synergistically Induces a Durable Antitumor Effect by Increasing Granzyme B-Secretion In Vivo

As in vitro functional studies showed the increased anticancer effect of CD8^+^ T cells on ovarian cancer cells after co-treatment with compound 968 and anti-PD-L1 antibody (Figure 3), we next explored whether similar effects would be seen in the xenograft model using C57BL/6J mice. Twenty-eight days after ID8 ovarian cancer cell inoculation, mice were randomly divided into four groups to receive compound 968 and anti-PD-L1 treatment alone or in combination (Figure 4A). Consistent with the in vitro assays, both compound 968 and anti-PD-L1 antibody could prolong the overall survival of mice, but combined treatment resulted in longer survival times than each treatment alone (median survival time 43.5, 64, 62, or >80 days for control, anti-PD-L1 antibody, compound 968, and combination treatment, respectively; Figure 4B). After euthanasia, significantly fewer tumors were disseminated on the surface of peritoneal organs, including peritoneum, gut, small intestine, and peritoneal cavity wall, in the combined treatment group compared with the anti-PD-L1 antibody group or the control (Figure 4C,D). The combined treatment group also displayed fewer tumors compared with the compound 968 group albeit statistical significance cannot be reached (*p* = 0.0958) (Figure 4D). The weights of tumors from mice in the combined treatment group were also significantly decreased compared with those from the control group (Figure 4E). 

Tumors excised from the mice in each group treatment were subjected to flow cytometry to detect granzyme B secretion by CD4^+^ and CD8^+^ T cells isolated from the tumors. The results showed that compound 968 could increase granzyme B secretion by CD4^+^ and CD8^+^ T cells. Importantly, tumor-bearing mice that received the combination treatment of compound 968 and anti-PD-L1 antibody had the highest granzyme B-producing cell fractions (Figure 5A–C). These results suggested that compound 968 could cooperate with anti-PD-L1 antibody to activate T cell killing and that this combination was more effective than the individual agents in the treatment of ovarian cancer.

### 3.6. Compound 968 Improves CD3^+^ T-Cell Infiltration into the Tumor Site 

The number of tumor infiltrated lymphocytes is thought to be the major component reflecting the anti-tumor immune response [32]. Studies also revealed the positive correlation between the number of CD3^+^ T cells and good clinical outcomes in ovarian cancer [33]. Therefore, we characterized the profiles of infiltrated CD3^+^ T cells into ID8 tumors by flow cytometry for each treatment group. Tumors were collected from the peritoneal cavity of mice after euthanasia and then underwent a dissociation procedure to obtain a single cell suspension. The flow cytometry results demonstrated that the CD3^+^ T cells (Figure 6A,B) infiltrated the tumors of mice treated with single compound 968 treatment had significantly higher numbers than the tumors of mice in the control group with a synergistic effect demonstrated with anti-PD-L1 antibody co-treatment although statistical significance cannot be reached (*p* = 0.0867). Taken together, our findings indicate that compound 968 can enhance the trafficking of immune cells into tumor sites in ovarian cancer.

### 3.7. Compound 968 Increases CXCL10 and CXCL11 Secretion by Cancer Cells 

T cell infiltration is a common feature of various types of cancer and is of prognostic and therapeutic relevance [34]. Furthermore, cytokines, including chemokines, play an essential role in regulating immune cell differentiation and infiltration. The chemokines CXCL10 and CXCL11 serve as chemo-attractants for dendritic cells, activated T lymphocytes, and natural killer cells that express the receptor C-X-C Motif Chemokine Receptor 3 (CXCR3) [35]. High expression of CXCL10 in ovarian cancer has been associated with improved overall survival [36]. Our in vivo data demonstrate compound 968 can increase CD3^+^ T cell infiltration into the tumor site (Figure 6). We further assessed the mRNA expression and secretion of CXCL10 and CXCL11 by ovarian cancer cells (ES2, A2780CP, OVCA433 or SKOV-3) after treatment with compound 968. The qPCR results revealed that mRNA expression of both of these chemokines increased upon treatment with compound 968 except that no change of CXCL11 was observed in A2780CP after compound 968 treatment (Figure 7A,B). Secretion of both chemokines were also enhanced in OVCA433 and SKOV-3 cells as determined by ELISA (Figure 7C,D). These results indicated that compound 968 could facilitate the infiltration of T cells, possibly through inducting the secretion of CXCL10 and CXCL11 by tumor cells.

## 4. Discussion

Immune checkpoint inhibitors, including CTLA4 and PD-1/PD-L1 antibodies, have revolutionized the management of tumors since their discovery. Although the United States Food and Drug Administration has approved the application of immune checkpoint inhibitors for the treatment of various types of tumors, up to two-thirds of patients suffer from primary or acquired resistance [37]. To address this clinical challenge, different treatment strategies have been tested in cooperation with immune checkpoint inhibitors to identify an attractive drug partner for therapy. In a recent clinical trial testing the safety and efficacy of nivolumab, a fully human anti-PD-1 antibody, in 20 ovarian cancer patients, the best overall response was 15% in the cohort receiving a high-dose of nivolumab intravenously including a complete response in two cases [8]. While in another clinical trial, the overall response was 11.5% in ovarian cancer patients treated with pembrolizumab, another highly selective humanized antibody target PD-1 [9]. Compared with the extraordinary effects on other types of cancer, such as melanoma and non-small cell lung cancer where immune checkpoint inhibitors have been reported to significantly prolong clinical survival, great efforts are being made to improve the efficacy of immune checkpoint inhibitors in ovarian cancer [31,38,39].

In this study, by analyzing the TCGA dataset, we found that the overexpression of GLS predicted worse progression-free survival and overall survival in ovarian cancer. Hudson and colleagues also identified a negative correlation between expression of GLS and prognostic outcomes (i.e., progression-free and overall survival) in ovarian cancer patients, and showed that inhibition of GLS activity by an inhibitor sensitized the ovarian cancer cells to cisplatin treatment [40]. An increased level of GLS was also reported to be associated with an advanced disease stage and poor prognosis in hepatocellular carcinoma [41]. Furthermore, we found that compound 968, as an inhibitor of glutaminase, was able to inhibit cell proliferation, which was consistent with the findings of Yuan and co-workers who reported that compound 968 inhibited cell proliferation and sensitized cells to paclitaxel treatment in ovarian cancer [19]. As a result of the high activity and expression of glutaminase, cancer cells present a higher rate of conversion of glutamine into glutamate [42]. Studies revealed that high concentration of glutamate in the tumor microenvironment may inhibit T cell activity has also been reported [43] and inhibition of glutamine metabolism increases the antitumor activity of CD8^+^ T cells [44]. In this study, analysis of the TCGA dataset revealed that GLS expression in ovarian cancer correlated with an immunosuppressive microenvironment. Taken together, our study and those of others proposed that glutaminase inhibition might be an ideal partner to improve the treatment effects of immunotherapy. 

Annexin V analysis of cancer cells isolated from both the direct and indirect co-culture systems with CD8^+^ T cells in the presence of compound 968 revealed an increased rate of apoptosis compared with other groups (without CD8^+^ T cells or compound 968). These results suggested that the antitumor efficacy by CD8^+^ T cells could be enhanced by the suppression of GLS activity with its inhibitor compound 968 in ovarian cancer. Similarly, Johnson and colleagues reported the increased expression of effector proteins, such as granzyme B and perforin, by CD8^+^ T cells through the inhibition of GLS [45]. These results indicated that inhibition of GLS helped to increase CD8^+^ T cell activity by triggering the secretion of effector molecules, and based on these findings, we investigated the cytokines or cytotoxic enzymes secreted by CD8^+^ T cells after exposure to compound 968. 

Interestingly, we found that compound 968 could enhance granzyme B secretion by CD8^+^ T cells isolated from PBMCs. After being released from cytotoxic T cells or natural killer cells, granzyme B would be taken up via receptor-mediated endocytosis by target cells, where it would initiate distinct apoptosis pathways to activate targeted cell death [46]. Inhibiting granzyme B by its natural inhibitor serpin B9 (PI-9) was reported to protect prostate cancer cells from natural killer cell-induced apoptosis [47]. Additionally, a patient with melanoma who showed a good response to treatment leading to the blockade of PD-1 showed increased mRNA expression of granzyme B and IFN-γ in the regressing tumor lesions [48]. In a syngeneic colon adenocarcinoma tumor model, anti-PD-L1 treatment reinvigorated CD8^+^ T cells by increasing granzyme B production [49]. In our study, we further confirmed the antitumor effect of CD8^+^ T cells following co-treatment with compound 968 with anti-PD-L1 antibody. Our results showed that the survival of ES2 cells significantly decreased in the co-culture system with CD8^+^ T cells treated with compound 968 and anti-PD-L1 antibody compared with the individual treatment groups. This may indicate that compound 968 can cooperate with the PD-1/PD-L1 blockade to enhance the T cell antitumor effect.

Similar to the nonimmunogenic human ovarian cancers, the ID8 tumors from the ovarian cancer mice model are also poorly infiltrated by T cells [50]. Besides, compared with early ID8 tumors, higher percentage of immunosuppressive immune cells, including Tregs, myeloid-derived suppressor cells (MDSCs), and tumor associated macrophages (TAMs) were found in advanced tumors [50]. We showed that the combination of compound 968 and anti-PD-L1 antibody elicited a synergistic antitumor effect in the ID8 ovarian cancer model. This combination treatment not only extended the overall survival of mice but also suppressed peritoneal tumor formation. Furthermore, granzyme B secretion by CD4^+^ and CD8^+^ T cells were higher in the tumor lesions of mice treated with compound 968 with anti-PD-L1 antibody compared with the other treatment groups. Consistent with the in vitro experiments, we demonstrated that compound 968 combined with anti-PD-L1 antibody had a beneficial therapeutic effect on ovarian cancer by increasing the cytolytic potential of immune cells.

Different immune backgrounds might have different effects on tumor progression, with higher numbers of cytotoxic T cells or lower numbers of immune suppressor cells correlating with a favorable prognosis in various types of cancer [51,52]. The assessment of immune cell composition in melanoma tumors revealed that the number of infiltrating CD8^+^ T cells predicted the response to anti-PD-1 therapy [53]. Therefore, it is crucial to clarify the mechanisms that modulate the infiltration of immune cells. A recent study revealed that loss of GLS in breast cancer cells not only reduced the tumor growth and metastasis, but also enhanced T cells activation and infiltration [54]. In this study, we also assessed the influence on T cells infiltration into tumor executed by compound 968.

After interacting with their receptors, chemokines transmit diverse signals to induce chemotaxis, angiogenesis, and cancer metastasis [55]. Among them, the CXCL10/CXCL11/CXCR3 axis has become a hot topic of research given its ability to mediate the differentiation and directional migration of leukocytes [56]. The levels of CXCL10 and CXCL11 were limited under homeostatic conditions but could be secreted by cancer cells, endotheliocytes, leukocytes, and fibroblasts in response to different stimulants. The CXCR3 ligand is mainly expressed on Th1-type CD4^+^ T cells, activated CD8^+^ T lymphocytes, and natural killer cells [57]. The CXCR3 knockout syngeneic murine model of B16 melanoma showed significantly accelerated tumor growth and reduced survival, along with reduced infiltration of CD8^+^ T cells into tumors [58]. Moreover, anti-PD-1 treatment in CXCR3 knockout mice with melanoma failed to reduce tumor growth, and this correlated with the reduced migration of T cells into tumors [58]. All of these results confirmed the essential role of T cell infiltration mediated by the CXCL10/CXCL11/CXCR3 axis in the success of PD1 blockade therapy. Additionally, maintaining the biologically active form of CXCL10 enhanced the recruitment of lymphocytes to tumor sites and improved the treatment effect of immune checkpoint inhibitors [59]. In this study, we found that the inhibition of GLS by compound 968 could increase the expression of CXCL10 and CXCL11 by cancer cells. We found a higher number of CD3^+^ T cells infiltrated into the tumors of mice treated with compound 968 compared with the control. These results suggested that compound 968 could also increase lymphocyte infiltration, thereby improving the therapeutic effect of immune checkpoint inhibitors. 

## 5. Conclusions

In this study, we report that compound 968 not only inhibited the proliferation of, but also augmented CXCL10 and CXCL11 secretion by ovarian cancer cells. Given the enhanced secretion of granzyme B by T cells and the increased infiltration of T cells into ID8 ovarian tumors following treatment with compound 968, we conclude that compound 968 is an ideal partner to cooperate with immune checkpoint inhibitors in ovarian cancer. Furthermore, we found that the anti-PD-L1 antibody could enhance the antitumor response when combined with glutaminase inhibitor compound 968 in the syngeneic murine model of ovarian cancer. Therefore, further studies are warranted to assess the safety and efficacy of this combination treatment before translating it to the clinic.

## Figures and Tables

**Figure 1 biomolecules-11-01749-f001:**
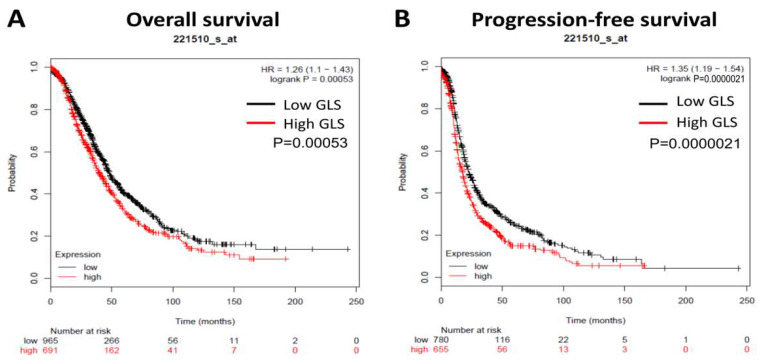
Upregulation of GLS predicts poorer overall/progression-free survival. Kaplan–Meier analysis usign the Kaplan–Meier plotter dataset from ovarian cancer patients (including all tumor stages, grades and histological subtypes) showed that GLS expression correlates with poorer overall survival (**A**) and progression-free survival (**B**).

**Figure 2 biomolecules-11-01749-f002:**
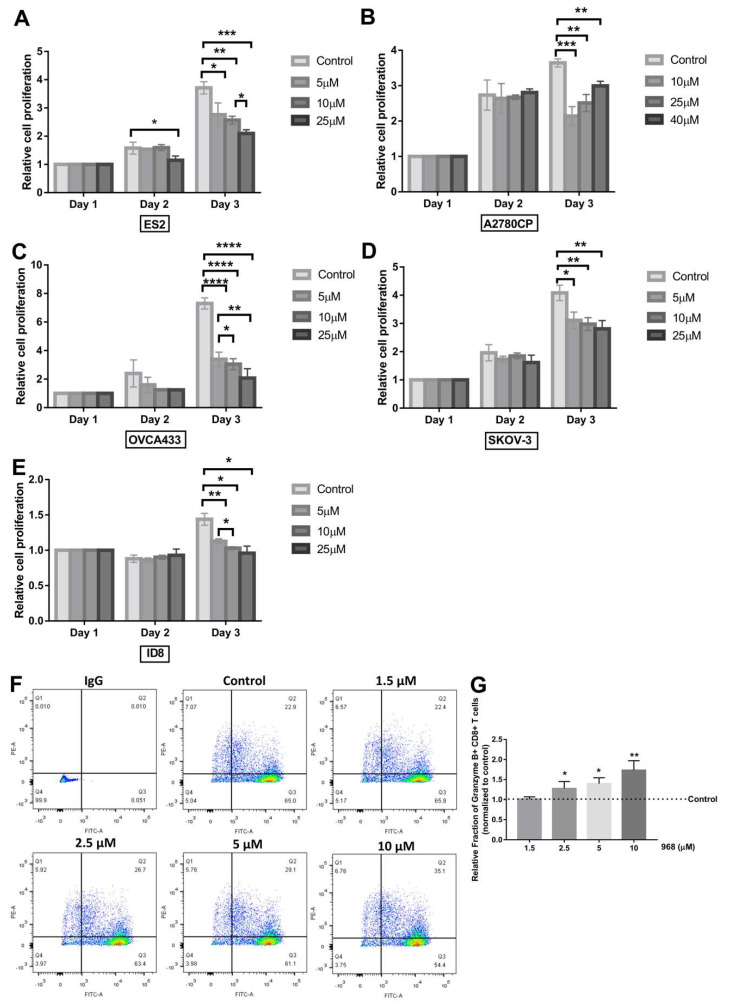
Compound 968 inhibited ovarian cancer cell proliferation and increased granzyme B secretion by CD8^+^ T cells. The human ovarian cancer cell lines ES2 (**A**), A2780CP (**B**), OVCA433 (**C**), SKOV–3 (**D**), and the mouse–derived ovarian cancer cell line ID8 (**E**) were treated with different concentrations of 968 for different times and XTT was used to examine the growth rate of cancer cells. (**F**,**G**) CD8^+^ T cells treated with different concentrations (0, 1.5, 2.5,5 and 10 μM) of 968 for 48 h were stained with antibodies against FITC–labeled CD8A and PE-labeled granzyme B. Representative contour showed granzyme B secretion by CD8^+^ T cells (**F**) and statistical analysis of these flow cytometry results (**G**). Representative data from three to five experiments are shown (* *p* < 0.05, ** *p* < 0.01, *** *p* < 0.001, **** *p* < 0.0001).

**Figure 3 biomolecules-11-01749-f003:**
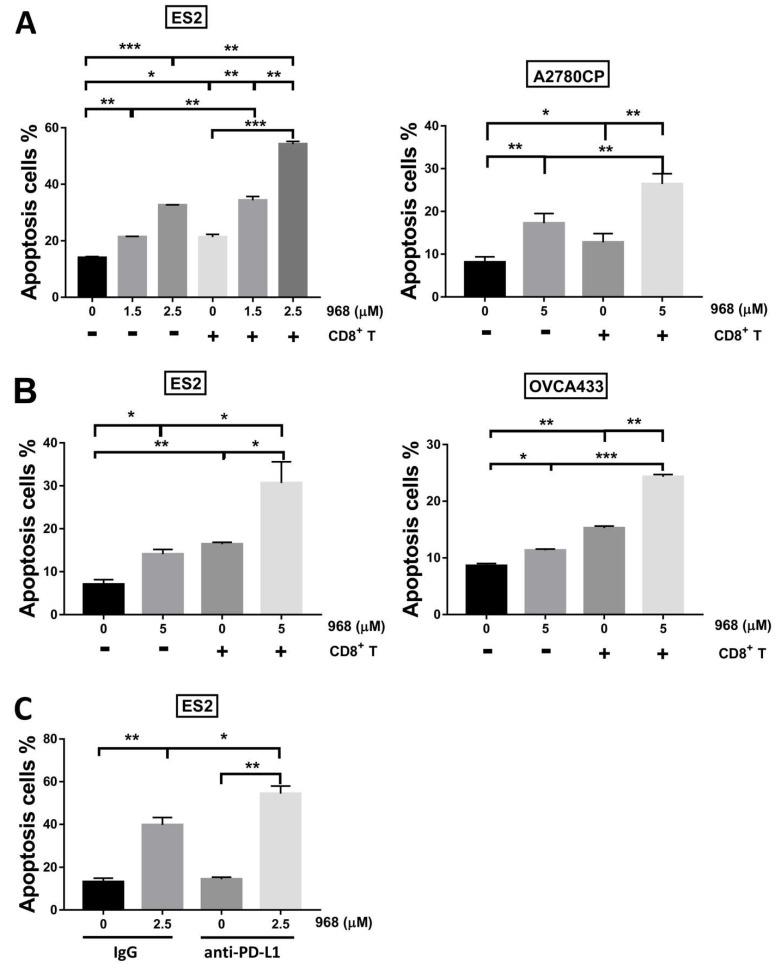
Apoptosis of cancer cells increased in the co–culture system with CD8^+^ T cells when exposed to compound 968. Flow cytometry was used to examine the apoptosis rate of cancer cells in the direct and indirect co–culture systems. (**A**) In the direct co–culture system, the apoptosis of ES2/A2780CP cells treated with different concentrations of 968 in the presence or absence of CD8^+^ T cells at a ratio of 1:5. The cancer cells and CD8^+^ T cells were collected at the end of co–culturing and were stained with FITC–annexin V and PE–anti–CD8 antibody. Annexin V+ cells within the CD8–gate were considered to be apoptotic cancer cells. (**B**) In the indirect co-culture system, apoptosis of ES2/OVCA433 cells seeded in the lower chamber treated with 0 or 5 μM of 968 with or without CD8^+^ T cells at a ratio of 1:5. (**C**) In the direct co–culture system, the apoptosis of ES2 cells increased when exposed to compound 968 and anti–PD–L1 antibody. The results are presented as the mean ± SD, and data from three experiments are shown (* *p* < 0.05, ** *p* < 0.01, *** *p* < 0.001).

**Figure 4 biomolecules-11-01749-f004:**
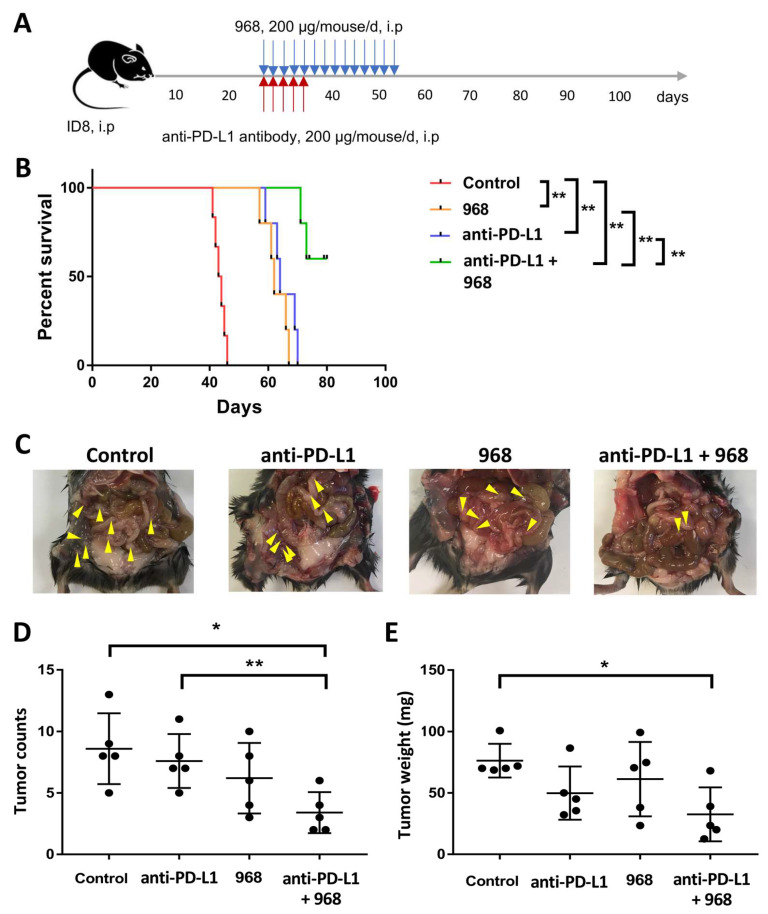
Treatment with glutaminase inhibitor 968 and anti-PD–L1 antibody increased overall survival in the ID8 ovarian cancer model. (**A**) The scheme of therapeutic. (**B**) Overall survival of mice treated with glutaminase inhibitor 968 and/or anti–PD–L1 antibody. ** *p* < 0.01. (**C**) The typical presentation of ID8 ovarian cancer in C57BL/6 mice treated with glutaminase inhibitor 968 and/or anti–PD–L1 antibody. (**D**,**E**) The peritoneal tumors were counted and weighed after euthanization. Each dot represents one mouse (* *p* < 0.05, ** *p* < 0.01).

**Figure 5 biomolecules-11-01749-f005:**
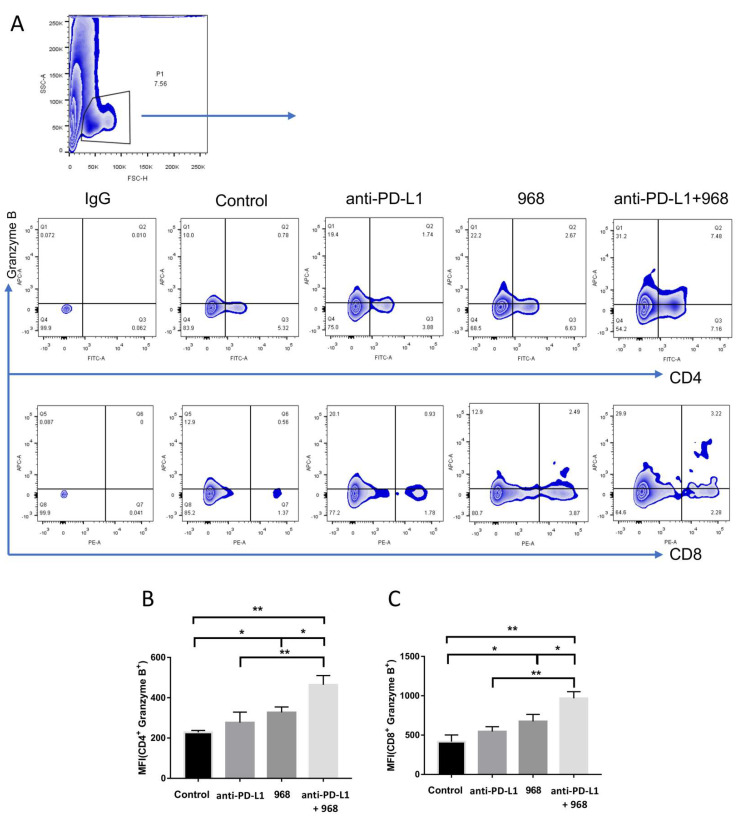
The combination of glutaminase inhibitor 968 with anti–PD–L1 antibody increased the granzyme B secretion of CD4^+^ T cells and CD8^+^ T cells that isolated from tumors. (**A**) Representative results of granzyme B–positive cells in CD4^+^ T cells (upper layer) and CD8^+^ T cells (lower layer). (**B**,**C**) Statistical analysis of granzyme B–positive CD4^+^ T cells (B) and granzyme B–positive CD8^+^ T cells (**C**) that infiltrated into tumors from the four groups. (* *p* < 0.05, ** *p* < 0.01).

**Figure 6 biomolecules-11-01749-f006:**
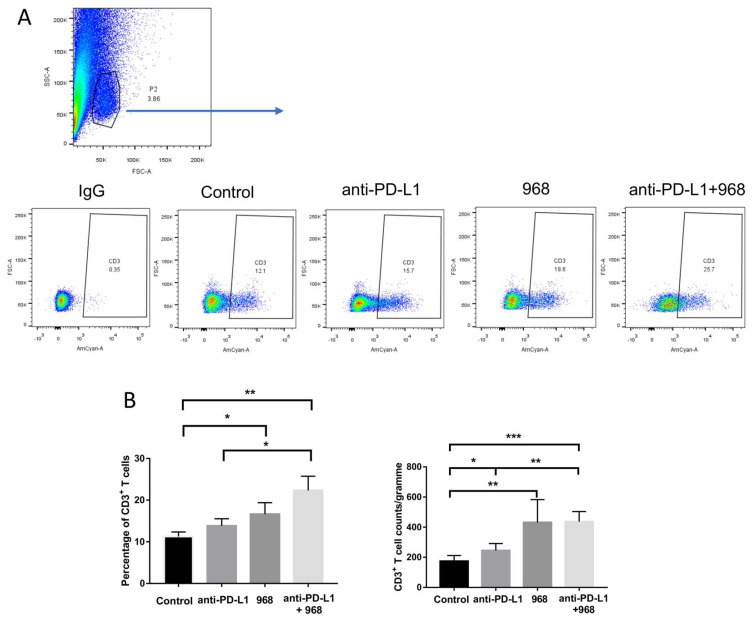
The combination of glutaminase inhibitor 968 with anti–PD–L1 antibody enhanced the infiltration of CD3^+^ T cells into tumors. (**A**) Representative results of CD3^+^ T cells infiltrated into tumors. (**B**) Statistical analysis of CD3^+^ T cells that infiltrated into tumors from the four groups (* *p* < 0.05, ** *p* < 0.01).

**Figure 7 biomolecules-11-01749-f007:**
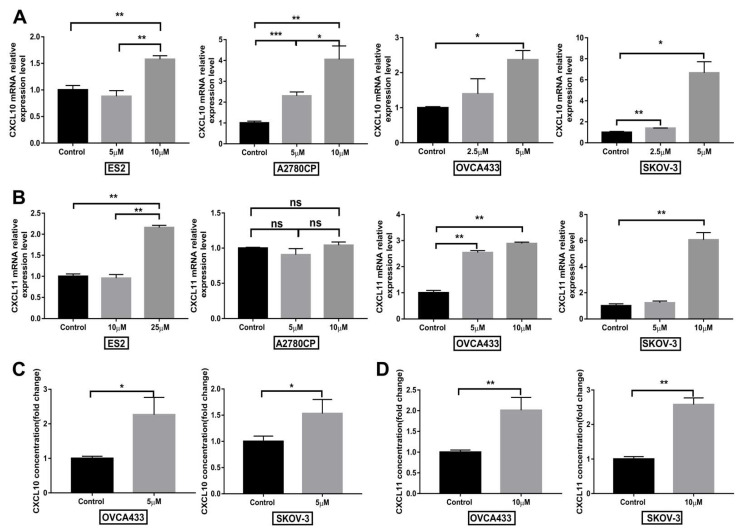
Compound 968 treatment increased CXCL10 and CXCL11 secretion by cancer cells. The human ovarian cancer cells (ES2, A2780CP, OVCA433 or SKOV-3) were treated with different doses of 968 for 48 h and q–PCR was used to determine CXCL10 (**A**) and CXCL11 (**B**) expression, ELISA was used to determine CXCL10 (**C**) and CXCL11 (**D**) secretion. Results are presented as the mean ± SD, and representative data from three experiments are shown (* *p* < 0.05, ** *p* < 0.01, *** *p* < 0.001).

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
