# Peer review of "A Combination of Glutaminase Inhibitor 968 and PD-L1 Blockade Boosts the Immune Response against Ovarian Cancer"

_biomolecules, 2021, doi:10.3390/biom11121749_

Round 1

Reviewer 1 Report

Summary: In this study, the authors investigate the potential combination of GLS inhibitor 968 with immune checkpoint inhibitors. While the concept is not novel and has been previously described in other tumor types, not all models may be suitable for this approach.

Here are specific concerns in this study:

  1. Figure 1: The authors try to correlate the GLS expression with various checkpoint markers. However, the correlation value of R is very low and is not convincing that a GLS high tumor may also have high PD-L1 expression etc. Further, given the heterogeneity in expression, this correlation should have been presented at the protein level. At the RNA level, Figure 1C is very weak and does not convince the reader of the study ahead.
  2. Re: the ID8 model- the authors do not describe the general immune milieu of this system. Is this model immunogenic at baseline? How does the immune profile look in the absence of any drugs.
  3. The authors show that there increased Granzyme B secretion in vitro on treatment with 968. How is this possible? Given the doses used, is it true secretion or merely cell death in T-cells. Assuming that 968 is a specific GLS inhibitor, the doses used in all in vitro studies are very high. A good dose response curve (esp. in tumor cells) is missing. Thus, the results presented seem to be resulting from off-target effects.
  4. Figure 5 starts to make sense. However, the authors need to give a complete picture of the immune profile. Showing absolute counts of each population is critical, as the % can be misleading in biological relevance if the number of cells infiltrating the tumor microenvironment is very small.
  5. Does the combination have an impact on metastasis? Does the model metastasize? How does the overall survival in mice compare when given standard of care?

Overall, the authors show a proof-of-concept study, the idea of which is not very novel in the field. The granzyme B secretion studies are problematic and not convincing of a valid synergy here.

Author Response

Reviewer 1:

Comment 1: Figure 1: The authors try to correlate the GLS expression with various checkpoint markers. However, the correlation value of R is very low and is not convincing that a GLS high tumor may also have high PD-L1 expression etc. Further, given the heterogeneity in expression, this correlation should have been presented at the protein level. At the RNA level, Figure 1C is very weak and does not convince the reader of the study ahead.

Response to Comment 1:

Thanks for this advance. We performed the correlation study using TCGA ovarian cancer dataset to show the potential effects of GLS on immunosuppressive microenvironment. To support this descriptive bioinformatic data, we performed functional studies (Figures 2-7) to reveal that the inhibition of glutaminase by compound 968 helped to improve the treatment effect of immune checkpoint blockade in ovarian cancer. We deleted the term “statistically” and moved this to Supplementary Figure S1 in Results section, line 233-236, page 5 as suggested by reviewer 2’s comment 3.

Comment 2: Re: the ID8 model- the authors do not describe the general immune milieu of this system. Is this model immunogenic at baseline? How does the immune profile look in the absence of any drugs.

Response to Comment 2: We appreciate the thoughtful comment of the reviewer. The immunocompetent C57BL/6J mice were used in this study. “Similar to the nonimmunogenic human ovarian cancers, the ID8 tumors from the  ovarian cancer mice model are also poorly infiltrated by T cells [31]. Besides, compared with early ID8 tumors, higher percentage of immunosuppressive immune cells, including Tregs,myeloid-derived suppressor cells (MDSCs), and tumor associated macrophages (TAMs) were found in advanced tumors [31].” Such information has been added in Discussion section, line 463-467, page 16.  

Comment 3: The authors show that there increased Granzyme B secretion in vitro on treatment with 968. How is this possible? Given the doses used, is it true secretion or merely cell death in T-cells. Assuming that 968 is a specific GLS inhibitor, the doses used in all in vitro studies are very high. A good dose response curve (esp. in tumor cells) is missing. Thus, the results presented seem to be resulting from off-target effects.

Response to Comment 3: Doses we used to treat cancer cells were based on a previous study [19]. Such information has been added in Materials and Methodssection, lines 155-156, page 4. Dose dependent effect on cell proliferation in ovarian cancer cells treated with compound 968 was shown in Figure 2.

Comment 4: Figure 5 starts to make sense. However, the authors need to give a complete picture of the immune profile. Showing absolute counts of each population is critical, as the % can be misleading in biological relevance if the number of cells infiltrating the tumor microenvironment is very small.

Response to Comment 4: Thanks for this suggestion. We have amended Figure 5 accordingly (Figures 5 and 6 in revised version).

Comment 5: Does the combination have an impact on metastasis? Does the model metastasize? How does the overall survival in mice compare when given standard of care?

Response to Comment 5: Similar to the findings reported by Hye Jeong Lee, et al., we observed tumors dissemination on the surface of peritoneal organs, including peritoneum, gut, small intestine, and peritoneal cavity wall in this ovarian cancer mice model. Moreover, fewer tumors were disseminated in the peritoneal cavity of mice in the group of combined treatment compared with the anti-PD-L1 antibody group or the control group. Such information has been amended in the Results section, line 327-330, page 10.   

Lee, H.J.; Tantawy, M.N.; Nam, K.T.; Choi, I.; Peterson, T.E.; Price, R.R. Evaluation of an intraperitoneal ovarian cancer syngeneic mouse model using 18f-fdg micropet imaging. Int J Gynecol Cancer 2011, 21, 22-27.

To date, the debulking surgery combined with chemotherapy have been the standard therapies for ovarian cancer patients. Studies have showed that the overall survival of C57BL/6 mice bearing ID8 tumors treated with anti-PD-L1 antibody was shorter than those treated with carboplatin [6] or paclitaxel [7] alone. Such information has been amended in the Introduction section, line 50-52, page 2.

Reviewer 2 Report

This is an interesting study. The use of therapeutic substances capable of potentiating each other's activity, while having non-synergistic side effects, is a very productive approach.

I have a number of questions and comments.

Line 207: monocytes

I suggest you separate not monocytes but mononuclear cells. In methods you a write about peripheral blood mononuclear cells (PBMCs).

Figure 5E: The Y-axis sign as Percentage of CD3+ T cells in monocytes of tumor. This is fundamentally wrong.

Line 234 and other: statistically significant correlation was observed between high expression of GLS and

Using term “statistically” not necessary.

Line 277: systems with CD8+ T cells at a ratio of 1:5 with or without compound 968 using the

Why do you use 1:5 ratio?

Line 292-293: FITC-annexin V or PE-anti-CD8 antibody. Annexin V+ cells within the CD8Ë— gate were considered to be apoptotic cancer cells.

How do you had possibilities to evaluate apoptotic cancer cells in CD8Ë— gate if you use flow cytometry with FITC-annexin V or PE-anti-CD8 antibody? You should use doble-color flow cytometry I suggest.

In the Figure 3C you represent data about apoptotic cells rate incubated with various condition, and it clearly shown that IgG exposure increase percent of apoptotic cells. What could be the reason for this?

Line 298: 3.4. Combination of compound 968 and anti-PD-L1 antibody enhances the anticancer activity of CD8+ T cells

Does this part perform in co-culture system?

Author Response

Reviewer 2:

Comment 1: I suggest you separate not monocytes but mononuclear cells. In methods you a write about peripheral blood mononuclear cells (PBMCs).

Response to Comment 1: Thanks for the advice. We have amended them accordingly. “Thereafter, the enriched mononuclear cells were stained with BV510-anti-CD3e (BD, #563024), FITC-anti-CD4 (eBioscienceTM, #553729), or PE-anti-CD8A according to the protocols provided by the suppliers.” Such information has been amended in Materials and Methods section, lines 207-210, page 5.

Comment 2: Figure 5E: The Y-axis sign as Percentage of CD3+ T cells in monocytes of tumor. This is fundamentally wrong.

Response to Comment 2: Appreciate for the considerate advice. Figure 5E: The Y-axis sign has been amended to Percentage of CD3+ T cells (Please see Figure 6B in revised version).

Comment 3: Line 234 and other: statistically significant correlation was observed between high expression of GLS and Using term “statistically” not necessary.

Response to Comment 3: The term “statistically” has been deleted in Results section, line 233-236, page 5.

Comment 4: Line 277: systems with CD8+ T cells at a ratio of 1:5 with or without compound 968 using the

Why do you use 1:5 ratio?

Response to Comment 4: We followed the study reported by Kota Iwahori, in which they checked the cytotoxic activity of tumor-infiltrating T cells by co-culturing the cancer cells and T cells at a ratio of 1:5. Besides that, Veronica Olivo Pimentel et al. also measured the viability of tumor cells after co-culturing with T cells at the ratio of 1:5 [21, 22]. Such references have been added to Materials and Methods section, line 128-131, page 3 .

Comment 5: Line 292-293: FITC-annexin V or PE-anti-CD8 antibody. Annexin V+ cells within the CD8Ë— gate were considered to be apoptotic cancer cells.

How do you had possibilities to evaluate apoptotic cancer cells in CD8Ë— gate if you use flow cytometry with FITC-annexin V or PE-anti-CD8 antibody? You should use doble-color flow cytometry I suggest.

Response to Comment 5: Based on the study performed by HaiDong Dong, et al.[23], we followed the method that examined the apoptotic cancer cells in CD8(-) gate by flow cytometry after staining the cells with FITC-annexin V and PE-anti-CD8 antibody. Such reference has been added to Materials and Methods section, line 171-172, page 4.

Comment 6: In the Figure 3C you represent data about apoptotic cells rate incubated with various condition, and it clearly shown that IgG exposure increase percent of apoptotic cells. What could be the reason for this?

Response to Comment 6: In the Figure 3C, compound 968 treatment (2nd lane: compound 968 and IgG) showed increased percentage of apoptosis cells when compared with IgG alone (1st lane: IgG), suggesting that compound 968 was able to induce apoptosis. We have amended the text. “The functional assay results showed that the percentage of apoptotic ES2 cells in the combination treatment group (compound 968 and anti-PDL1 antibody) increased by about 15% compared with the group treated with compound 968 alone (compound 968 and IgG), and by approximately 40% compared with the group treated with anti-PD-L1 antibody alone (Figure 3C). This finding confirmed that the combination of compound 968 with PD-L1 antibody could be an effective co-treatment method to improve the clinical effects of immunotherapy.” Such information has been amended in Results section, lines 308-312, page 10.

Comment 7: Line 298: 3.4. Combination of compound 968 and anti-PD-L1 antibody enhances the anticancer activity of CD8+ T cells

Does this part perform in co-culture system?

Response to Comment 7: Yes, this part was performed in co-culture system.

Round 2

Reviewer 2 Report

Comment 5: Line 292-293: FITC-annexin V or PE-anti-CD8 antibody. Annexin V+ cells within the CD8Ë— gate were considered to be apoptotic cancer cells.

How do you had possibilities to evaluate apoptotic cancer cells in CD8Ë— gate if you use flow cytometry with FITC-annexin V or PE-anti-CD8 antibody? You should use doble-color flow cytometry I suggest.

Response to Comment 5: Based on the study performed by HaiDong Dong, et al.[23], we followed the method that examined the apoptotic cancer cells in CD8(-) gate by flow cytometry after staining the cells with FITC-annexin V and PE-anti-CD8 antibody. Such reference has been added to Materials and Methods section, line 171-172, page 4.

If you perform double staining test to count apoptotic cells in CD8- gate you should say "and" as you do it in your response (FITC-annexin V and PE-anti-CD8 antibody), not "or".

Author Response

Reviewer 2:

Comment 5: Line 292-293: FITC-annexin V or PE-anti-CD8 antibody. Annexin V+ cells within the CD8Ë— gate were considered to be apoptotic cancer cells.

How do you had possibilities to evaluate apoptotic cancer cells in CD8Ë— gate if you use flow cytometry with FITC-annexin V or PE-anti-CD8 antibody? You should use doble-color flow cytometry I suggest.

Response to Comment 5: Based on the study performed by HaiDong Dong, et al.[23], we followed the method that examined the apoptotic cancer cells in CD8(-) gate by flow cytometry after staining the cells with FITC-annexin V and PE-anti-CD8 antibody. Such reference has been added to Materials and Methods section, line 171-172, page 4.

Reviewer’s comment: If you perform double staining test to count apoptotic cells in CD8- gate you should say "and" as you do it in your response (FITC-annexin V and PE-anti-CD8 antibody), not "or".

Response to reviewer’s comment: We appreciate the thoughtful comment of the reviewer. “The cancer cells and CD8+ T cells were collected at the end of co-culturing and were stained with FITC-annexin V and PE-anti-CD8 antibody.” Such information has been amended in legend of Figure 3, line 291-292, page 9.